# Identification of Eye Diseases Through Deep Learning

**DOI:** 10.3390/diagnostics15070916

**Published:** 2025-04-02

**Authors:** Elena Acevedo, Dinora Orantes, Marco Acevedo, Ricardo Carreño

**Affiliations:** 1Instituto Politécnico Nacional, Escuela Superior de Ingeniería Mecánica y Eléctrica, Campus Zacatenco, Mexico City 07700, Mexico; macevedo@ipn.mx; 2Instituto Politécnico Nacional, Centro de Investigación en Computación, Mexico City 07700, Mexico; sorantesj@ipn.mx; 3Department of Systems Engineering, Universidad del Istmo, Campus Tehuantepec, Ciudad Universitaria S/N, Barrio Santa Cruz, 4a. Sección Sto. Domingo Tehuantepec, Oaxaca 70760, Mexico; ricardo.carreno.a@sandunga.unistmo.edu.mx

**Keywords:** artificial intelligence, deep learning, convolutional neural networks, classification, eye diseases

## Abstract

**Background**: Ocular diseases have been a severe problem worldwide, specifically in underdeveloped countries that do not have enough technology or economy to treat them. It would be beneficial to have software with low installation complexity and ease of use, allowing high efficacy in diagnosing eye diseases. This study aims to design and implement an algorithm based on deep learning to classify ocular diseases with high precision. **Methods**: This work describes digital image processing techniques for the easier handling of eye images; in particular, blur filters were used. The Canny filter was also applied to obtain the edges that allow the difference between the analyzed diseases. Once the images were pre-processed, a convolutional neural network of our own design was applied to perform the classification task. The validation algorithm used in this work was the hold-out algorithm (80–20). The metrics used to evaluate our proposal were the confusion matrix, accuracy, recall precision, and F1-score. **Results**: The dataset has five classes, namely, normal, cataract, diabetic retinopathy, glaucoma, and other retina diseases. The network architecture consists of 11 layers, including three convolutional layers, three max pooling layers, one batch normalization layer, one flattening layer, two hidden layers, and one output layer. This model resulted in 97% efficiency across all metrics. **Conclusions**: With the individual analysis of each metric, it can be observed that the proposed algorithm is capable of differentiating, first, images of healthy eyes from diseased ones and, second, adequately classifying eye diseases.

## 1. Introduction

Retinopathy and glaucoma are diseases that primarily affect the population of Mexico; it is estimated that there are about 1.5 million cases of Glaucoma, and up to 50,000 cases of blindness, as there is no early detection system [1].

Glaucoma and diabetic retinopathy, along with other eye diseases, are a significant problem that must be addressed early to prevent blindness. These diseases have been a severe problem worldwide, specifically in underdeveloped countries that do not have enough technology or economy to treat them. Therefore, there is a need for pattern recognition algorithms that can help diagnose eye diseases economically and with adequate efficiency.

The basic definition of some eye diseases is described below.

Glaucomas is a group of optic neuropathies characterized by progressive degeneration of retinal ganglion cells. These are central nervous system neurons that have their cell bodies in the inner retina and axons in the optic nerve. The degeneration of these nerves results in cupping, a characteristic appearance of the optic disc and visual loss [2].

Diabetic retinopathy [3] occurs, because high blood sugar levels cause damage to the blood vessels in the retina. These blood vessels can swell and leak fluid and even close and prevent blood from flowing. Sometimes, it generates new abnormal blood vessels in the retina, causing vision loss.

A Cataract [4] is when the natural lens of the eye becomes cloudy. Proteins in your lens break down and make things look blurry, fuzzy, or less colorful.

Artificial intelligence (AI) [5] is the branch of computer science that makes computers mimic human behavior to assist humans for better performance in science and technology.

One of the primary advantages of AI is its ability to process and analyze vast amounts of data at unprecedented speeds, enabling informed decision making in sectors ranging from healthcare to finance. In the case of healthcare, different types of AI algorithms can be found to diagnose many diseases. Some examples are mentioned below.

Different artificial neural networks have diagnosed breast cancer, namely, quantum classical hybrid convolutional neural networks [6], convolutional neural networks [7,8], and graph neural networks [9].

Convolutional neural networks [10,11] and transfer learning algorithms [12] have been applied to diagnose Alzheimer’s disease. These algorithms have also been used in the diagnosis of pneumonia [13,14] and for the classification of skin diseases [15,16,17], among many other areas of medicine.

On the other hand, artificial intelligence is revolutionizing the diagnosis and treatment of eye diseases. With the increasing prevalence of conditions such as the diseases described above, there is a pressing need for innovative solutions that can enhance early detection, prognosis, and personalized treatment plans. AI technologies, including machine learning and neural networks, are being integrated into imaging techniques, such as fundus photography and optical coherence tomography, to analyze ocular data with remarkable precision. AI systems can facilitate the timely diagnosis of these diseases, potentially preserving vision and improving the quality of life of patients by processing vast amounts of data and identifying patterns that may elude even seasoned ophthalmologists.

Some works related to our proposal are described below.

Dheeraj and Ghosh [18] classified the following six ocular diseases: age-related macular degeneration, cataracts, diabetes, glaucoma, hypertension, and myopia. They applied a vision transformer-based approach with three different architectures with 8, 14, and 24 layers. The architecture with 14 layers obtained the best results with an F1-score of 83.49%, 84% sensitivity, 83% precision, and 0.802 Kappa score. They used the ODIR dataset, trained and validated the algorithms with 80% of the dataset, and used the 20% for testing.

Wahab [19] built a deep learning-based eye disease classification model. Three steps are applied; the first was detecting the main features using the single-shot detection algorithm. Then, the whale optimization algorithm (WOA) with levy flight and wavelet search strategy was applied for feature selection, and finally, the ShuffleNet V2 model was used for image classification. The following eight classes are analyzed: normal, DR, glaucoma, cataracts, age-related macular degeneration, hypertension, pathological myopia, and other diseases/abnormalities. The following two datasets are used: the ocular disease intelligent recognition (ODIR) dataset and the EDC dataset that was used to train the ShuffleNet V2 model. Both datasets were obtained from the Kaggle platform. The dataset was split into two parts, 70% for training and 30% for testing. The metrics and their corresponding results were accuracy = 99.1%, precision = 98.9%, recall = 99%, F1-score = 98.9%, Kappa = 96.4%, sensitivity = 98.9%, and specificity = 96.3%.

In 2023 [20], the authors applied three deep learning-based approaches, namely, the EfficientNetB0, VGG-16, and VGG-19 models. The dataset was obtained from the Kaggle platform, which contains images of the following four eye diseases: normal, diabetic retinopathy, cataract, and glaucoma. The dataset was split into training and testing, 70% and 30%, respectively. The metrics used to measure the performance of the classification models were accuracy, precision, recall, and AUC. The algorithm that obtained the best results was EfficientNetB0, with accuracy = 98.47%, precision = 96.98%, recall = 96.91%, and AUC = 99.84%.

Mostafa et al. [21] used a convolutional neural network with a pre-processing of the normalization and CLAHE algorithm. They analyzed six distinct diseases, including glaucoma, cataract, diabetes, age-related macular degeneration, hypertension, pathological myopia, and other diseases not explicitly mentioned in the Ophthalmic Disease Recognition (ODIR) dataset. The dataset was split into two groups for 70% training and 30% validation. The authors applied a data augmentation algorithm to balance the classes. The metrics used to measure the performance of the classification models were accuracy, precision, recall, and AUC. Two experiments were developed; the first consisted of classifying all the diseases together (multiclass classification), and in the second, each of the diseases was taken and classified together with the normal class (binary classification). In the first experiment, they obtained an accuracy of 60.31% and an AUC of 85%. For the second experiment, the accuracy was between 98% and 100%, recall from 97.99% to 100%, and precision between 96% and 100%.

Two classification models were used to detect eye diseases, namely, a convolutional neural network and a pre-trained model, EfficientNet CNN [22]. The diseases to be classified are cataract, diabetic retinopathy, glaucoma, and normal. The complete dataset combines four other sets with around 4200 colored images. The data were split into the following three subsets 70% training, 20% testing, and 10% validation. The following four metrics were applied precision, recall, F1-score, and accuracy. The best accuracy was obtained with the EfficientNet CNN with 94%.

Elkholy and Marzouk [23] used a pre-trained convolutional neural network with fine-tuning to detect three eye diseases by analyzing retinal images. They used the images from the dataset Optical Coherence Tomography (OCT). The dataset is balanced. The four classes were normal retina, diabetic macular edema (DME), choroidal neovascular membranes (CNM), and age-related macular degeneration (AMD). All classes have 8867 images. The images were pre-processed, enhanced, and restored. These images were fed into a VGG-16 convolutional neural network that classifies the eye diseases. The model accuracy was about 94%, and it reached 97% after fine-tuning.

A convolutional neural network was implemented to classify three types of eye diseases, namely, choroidal neovascularization, diabetic macular edema, and drusen [24]. The dataset is obtained from the Kaggle platform and has four classes, the three types of diseases and images of healthy eyes. The four classes add 84,495 retinal Optical Coherence Tomography (OCT) images. The dataset is split into three subsets, training (99%), testing (0.97%), and validation (0.03%). Three pre-trained models were applied to extract features, namely, VGG-16, Xception, and MobileNet. A convolutional neural network was used for classification. The CNN has two hidden layers; the first has 516 neurons, and the second has 216 neurons. The MobileNet with CNN ensemble model achieves the highest average accuracy, precision, recall, and f1-score, with an average accuracy of 95.34%.

In the current year, Al-Fahdawi et al. [25] proposed a Fundus-DeepNet system to classify eight ocular diseases. The fundus images were taken from the OIA-ODIR dataset. The dataset comprises 10,000 fundus images representing eight distinct ocular diseases, namely, normal case, diabetic retinopathy, glaucoma, cataracts, AMD, myopia, hypertension, and other abnormalities. The images had a pre-processing procedure, including circular border cropping, image resizing, contrast enhancement, noise removal, and data augmentation. The left and right fundus images are fed to a backbone network to extract the global features from the pre-processed images. The attention block learns additional high-level feature representations to differentiate lesion portions using the output of the backbone network. The fusion process is implemented two times in the attention block and three times in the SENet block. A discriminative restricted Boltzmann machine performs the task of classification. The Fundus-DeepNet system demonstrated F1-scores, Kappa scores, AUC, and final scores of 88.56%, 88.92%, 99.76%, and 92.41% in the off-site test set and 89.13%, 88.98%, 99.86%, and 92.66% in the on-site test set.

It can be observed that artificial intelligence tools are appropriate for the diagnosis of eye diseases. It is also worth highlighting that it is vitally important to obtain an adequate result in terms of precision and effectiveness, since it involves evaluating a person’s eye. Therefore, the proposed algorithm must present improvements in the results of both metrics. In this work, a convolutional neural network of our own design is proposed, and adequate results are presented in classifying eye diseases.

Our main contribution is an own-designed CNN, with a less complex architecture than a pre-trained algorithm, which allows, firstly, to differentiate a healthy eye from a diseased eye effectively and, secondly, to classify four eye diseases, such as cataract, diabetic retinopathy, glaucoma, and other retina diseases. All this is carried out through pre-processing, using blur filters and the Canny edge detector, which allows for better detection of main features and, consequently, a good performance of the CNN.

This paper is organized as follows. Section 2 briefly describes the algorithms used to perform the classification of eye diseases, namely, pre-processing and convolutional neural networks, along with the metrics used to test the performance of the proposed classification model. This section also describes the proposed convolutional neural network. Section 3 presents the results of the tests applied to our proposal. Section 4 reviews and comments on the results obtained. Finally, Section 5 presents the conclusions.

## 2. Materials and Methods

In this section, the algorithms used for both pre-processing and classification of eye diseases will be described. First, the filters used will be described, and then, the architecture of the convolutional neural network that performs the classification task will be shown. The methodology used to carry out the classification will then be presented. The dataset, the pre-processing algorithms, and the architecture of the convolutional neural network are described. Finally, the metrics used to evaluate the proposed model are shown.

### 2.1. Filters

Blur filters [26] are used to soften images or selections and are helpful for retouching. This filter reduces sudden changes in light intensity and, therefore, the contrast of images. The most visible consequence is that it blurs images, which helps eliminate noise.

There are some types of these filters, which are described below.

Box blur: The filter distorts the input image in the following way. Every pixel *x* in the output image has a value equal to the average value of the pixel values from the 3 × 3 square that has its center at *x*, and the value of *x* is included. Finally, all the pixels on the border of *x* are then removed.

Gaussian blur: In this case, the value of each pixel in the target image is a weighted Gaussian mean of the contents within the kernel in the source image. It effectively removes Gaussian noise from an image, which appears as a random variation in brightness or color.

Median blur: With this filter, the pixels of the target image are generated by calculating the median of those under the kernel placed above the corresponding ones of the source image. This type of filter works very well when the noise of the image is random.

Bilateral blur: A bilateral filter that is non-linear, edge-preserving, and noise-reducing, and it smooths images. It replaces the intensity of each pixel with a weighted average of intensity values from nearby pixels.

Canny edge detector [27]: This filter is composed of three basic stages—noise filtering, gradient, and hysteresis. Edge detection is susceptible to noise in the image, so a Gaussian filter with a 5 × 5 kernel is applied. The gradient is used to detect the intensity differences in adjacent pixels; for example, the gradient will be maximum in the vicinity of a black line drawn on a soft background. Edge detection is carried out through hysteresis, establishing a maximum and a minimum threshold; any area with a gradient greater than the maximum threshold value will be an edge.

### 2.2. Convolutional Neural Networks

The architecture of a CNN [28] consists of several types of layers, as follows:

Input layer: It is the layer that receives each of the images in the dataset.

Convolutional layer: In this layer, the image is convolved with some kernel to obtain the main features of the image. The output of this layer is referred to as feature maps.

Activation layer: Nonlinearity is essential in neural networks, so that they can solve problems with patterns that are not linearly separable. That is why activation functions are applied to add nonlinearity to the networks. The rectified linear unit (RELU) function is typically used to achieve this goal.

Pooling layer: Convolution increases the original data, so reducing that amount of information is necessary. This layer reduces the dimension of the feature maps, making the process more efficient.

Flattening layer: After passing through the convolutional part of the network, it is necessary to flatten the data, that is, convert the feature maps to a one-dimensional vector. For this purpose, this layer is used to resize the convolution result.

Fully connected layers: At this stage, the classification task is carried out, where several hidden layers are involved, which are also activated, in general, with the RELU function.

Output layer: Finally, the output layer will be activated with different functions, depending on whether the classification is binary or multiclass.

### 2.3. Evaluation Metrics

A confusion matrix is a table with *n* combinations of predicted and actual values, where *n* is the number of classes. This matrix is called confusion, because the correctly and incorrectly classified patterns can be viewed as true positives, true negatives, false positives, and false negatives, as shown in Table 1.

The definitions of the terms from Table 1 are as follows:

True positives and true negatives: these are the counts of the correctly predicted data points in the positive and negative classes, respectively.

False positives: they are Type 1 errors and refer to the count of the data points that belong to the negative class but were predicted to be positive.

False negatives: they are Type 2 errors and refer to the count of the data points that belong to the positive class but were predicted to be negative.

Accuracy is the proportion of the correct predictions to the total number of predictions. Equation (1) shows the way to calculate this metric.(1)$$Accuracy=\frac{Number\;of\;correct\;predictions}{Total\;number\;of\;predictions}$$

Accuracy is useful when working with balanced classes, and it is concerned with the overall “correctness” of the model and not the ability to predict a specific class.

Precision indicates the general performance of the classifier. The Equation for precision is shown in Equation (2).(2)$$Precision=\frac{True\;positives}{True\;positives+False\;positives}$$

Precision works well for problems with imbalanced classes, as it shows the correctness of the model in identifying the target class. It is useful when the cost of a false positive is high. In this case, the target class has to be detected, even if some (or many) cases are missed, i.e., the patterns are classified as target class, but they belong to the other class.

Recall indicates the ability of the classifier to identify positive instances, as shown in Equation (3).(3)$$Recall=\frac{True\;positives}{True\;positives+False\;negatives}$$

It works well for problems with imbalanced classes, as it focuses on the ability of the model to find objects of the target class.

Recall is useful when the cost of false negatives is high. In this case, it typically wants to find all objects of the target class, even if this results in some false positives (predicting a positive when it is a negative).

F_1_-score is a metric that combines precision and recall into a single number that can be used for a fair judgment of the model and is equal to the harmonic mean of these two metrics. Equation (4) shows F_1_-score.(4)$$F_{1}\text{-}score=2\times \frac{Precision\times Recall}{Precision+Recall}$$

The value of F_1_-score is in the range of 0 and 1. When the value is 0, precision or recall can be 0; conversely, if F_1_-score is 1, precision and recall can be 1. Therefore, if the value is close to 1, the classifier will perform better.

### 2.4. Proposed Model

Figure 1 shows the flowchart illustrating the process for ocular disease classification.

The input image is pre-processed to obtain a clear, noise-free image, and an edge detector is applied to enhance the characteristics of each disease. These features are fed into a convolutional neural network to classify eye diseases.

### 2.5. Dataset

Two datasets were used. From the first dataset, called Cataract Prediction [29], the subset Retina_Disease was obtained; this subset does not include diabetic retinopathy. The second dataset was also obtained from Kaggle, and its name is Eye_Diseases_Classification [30]; in this dataset are the subsets normal, cataract, diabetic retinopathy, and glaucoma diseases. Therefore, there are five classes, each with 200 instances, which means the dataset is balanced.

In Figure 2, an example of the type of images that are handled in this work can be observed.

The two datasets were homogenized, and the dimensions of the resulting images were 250 × 250 pixels.

### 2.6. Pre-Processing

Images need to be cleaned to visualize their characteristics better and eliminate each impurity. This is achieved by applying various filters depending on what needs to be highlighted or eliminated. Therefore, the images were first changed to grayscale. The blur filter was used in a 3 × 3 matrix, followed by a median blur in a 5 × 5 matrix, and finally, a Gaussian filter in a 5 × 5 matrix to eliminate the most significant impurities and soften the image. This way, a sharper image could be obtained regarding contrasts, thus obtaining clearer features and the desired result. Finally, the Canny filter was applied to obtain the edges and features representing each disease.

In Figure 3, the results of the pre-processing step for the five classes are shown.

The images from Figure 3 are fed to the convolutional neural network to carry out the classification task.

### 2.7. Classification

The predictive model used is a custom convolutional neural network (CNN) using TensorFlow and Keras. The architecture of CNN is as follows.
✓The convolutional stage shows the following layers.There are three layers of convolution:Layer 1: 32 filters with a kernel of 5 × 5.Layer 2: 64 filters with a kernel of 3 × 3.Layer 3: 128 filters with a kernel of 3 × 3.✓The max pooling layers have kernels with dimension of 3 × 3.✓A batch normalization layer is included.✓The flatten layer delivers a vector of dimension 100 to the multilayer perceptron.✓In the multilayer perceptron stage, we proposed two hidden layers with 100 and 23 neurons, respectively, with ‘relu’ activation function and one output layer with ‘SoftMax’ activation function. Adam optimizer was applied. The loss function was categorical cross-entropy, and we proposed 50 epochs and a batch size of 50.

## 3. Results

The complete code was programmed in Python 3.11, using OpenCV libraries for image pre-processing. TensorFlow and Keras libraries were used to build the CNN, an open-source library written in the same language. The characteristics of the computer with which the program was codified were the following:8th generation Intel CORE i7 + processor at 2.2 GHz and Turbo Boost at 4GHz.NVIDIA GEFORCE GTX 1050 graphics card.8 GB RAM.Windows 10 Home operating system.

The validation method used was hold-out with 80% training and 20% testing.

The model was trained using the Adam optimization algorithm with a loss function primarily used for multiclass classification, and the accuracy metric was used to evaluate the performance of the model during training and validation. Early stopping callbacks were defined to prevent overfitting and restore the weights of the model that performed best on the validation set and model checkpoint to save the best-trained model. The model was trained using a data generator to increase the dataset size with a maximum of 50 epochs, and the validation set was used to monitor performance and prevent overfitting.

Effectiveness tests were performed using the images without any pre-processing, and the results are shown in Figure 4.

Figure 4 shows that the CNN model has good performance in terms of the loss function, but the accuracy shows a value of less than 1. One of the goals of this work is to improve the effectiveness of the results shown by related works. Therefore, applying the filters described in Section 2 was proposed to try to improve the results. Figure 5 shows the function lost and the accuracy results after applying the filters.

In Figure 5, it can be observed that, as in Figure 4, the graph of the loss function shows a good performance of the model, but in the case of accuracy, better results are shown when pre-processing is applied to the images. This is the reason why it was decided to use the blur and Canny filters to improve the results.

Below are the evaluation metrics that were applied to analyze the performance of our proposal with the application of the filters to the eye images.

Figure 6 shows the results of the confusion matrix.

Confusion Matrix Analysis:Class 0 (glaucoma): it is observed that, of the 33 instances of glaucoma, 32 were correctly classified, with only 1 instance misclassified as another condition.Class 1 (cataract): for the 48 instances of cataract, 47 were correctly classified, and 1 was incorrectly classified.Class 2 (retina): of the 37 instances of retinopathy, 36 were classified correctly, with 1 error.Class 3 (diabetic retinopathy): All 38 instances of diabetes were correctly classified, showing perfect performance in this class.Class 4 (normal): From the 44 normal instances, 43 were correctly classified, with only 1 error.

The confusion matrix shows us four cases in which the model fails in the classification. In one of the cases, it confuses class 0 with class 2. It also confuses class 0 with class 4, and there are cases in which it confuses two patterns of class 3 with class 2 and misclassifies class 4 and confuses it with class 1.

Table 2 shows the results of precision.

The interpretation of Table 2 is detailed as follows:

Macro: It is an arithmetic average of the accuracy of each class without considering class imbalance. A macro accuracy of 0.97218 indicates that, on average, the model has high accuracy for all classes.

Micro: This considers the total number of true positives and false positives at the global level. A micro precision of 0.97 shows that the overall performance of the model is very high.

Weighted: This is the weighted average of the precision of each class, considering the number of instances per class. This value is identical to micro, suggesting that the classes are balanced.

None: This is the individual precision for each class. The cataract, retina, and normal classes have lower accuracies (0.9591, 0.925, and 0.976, respectively), while the glaucoma and diabetes classes have perfect accuracy (1.0).

The accuracy results are presented in Table 3.

The results in Table 3 can be interpreted as follows:

Normalized accuracy, with an accuracy of 0.97, indicates that 97% of the predictions were correct.

Non-normalized accuracy indicates that 194 of the 200 instances in the test set were classified correctly.

Results from the recall metric are shown in Table 4.

Recall, shown in Table 4, measures the proportion of true positives over the total number of truly positive cases, as follows:

Macro: a value of 0.9701 shows that, on average, the model can identify all classes correctly.

Micro: a value of 0.97 indicates adequate overall performance of the model in correctly identifying positive cases.

Weighted: as with accuracy, the weighted value is 0.97, which suggests adequate overall performance.

None: individual recall shows that the model has the most difficulty with glaucoma, cataract, diabetes, and normal classes (0.9696, 0.9791, 0.9473, 0.95454, respectively), while it has a perfect recall for retina (1.0).

Finally, the results for F_1_-score are shown in Table 5.

The F_1_-score results in Table 5 show that:

Macro: a macro F_1_-score of 0.9706 indicates that the overall performance of the model is high and balanced across all classes.

Micro: a micro F_1_-score of 0.97 confirms the good overall performance of the model.

Weighted: a weighted F_1_-score of 0.97 suggests that the model performs well.

None: the individual values reflect the same trends observed in accuracy and recall, with lower performance in retinopathy and glaucoma.

## 4. Discussion

Figure 6, which shows the confusion matrix, indicates that the algorithm used in this work performs correctly, since it presents few erroneous data; the highest number of confusions is 2. In the case of precision (Table 2), it can be observed that the classes of glaucoma and diabetic retinopathy are correctly classified, while the other two diseases present an average precision of 0.94; however, the algorithm continues to show adequate performance when detecting if an image corresponds to a disease and differentiating it from a normal image. On the other hand, the recall metric tells us that the average obtained from the four diseases is 0.974. At the same time, the normal class shows a recall of 0.95, which, similarly to precision, indicates that the algorithm can differentiate a normal image from a diseased one. Overall, the results show us the following two issues: our proposal can correctly distinguish a healthy eye from a diseased one, and it can adequately classify the four eye diseases.

From the results presented in the previous section, it can be observed that the average obtained from all the metrics used is 0.97. The five classes to be analyzed were glaucoma, cataract, retina diseases, diabetic retinopathy, and normal. In this context, the works related to our proposal will be compared. Table 6 shows this comparison.

From Table 6, it can be observed that from the works that classify the diseases, including age-related macular degeneration, cataracts, diabetes, glaucoma, hypertension, and myopia [19,22,26], the one that uses the Fundus-DeepNet system [26] presents the best results with F1-score = 88.56% and AUC = 99.76%. It should be noted that the model that uses a CNN [22] presents an accuracy of up to 100%, but this is when it performs a binary classification and not a multiclass one.

The VGG-16 algorithm was used in [24,25] and applied to classify different types of eye diseases, obtaining an accuracy of 94% and 95.34%, respectively.

From the papers that handled four classes—glaucoma, cataract, diabetic retinopathy, and normal [20,21,23]—the ShuffleNet V2 model [20] obtained an accuracy of 99.1% being the best result of the three algorithms.

A direct comparison with any of the previous algorithms cannot be made, because the datasets are not the same. However, it can be observed that our proposal presents 2.1% less accuracy than the ShuffleNet V2 algorithm. Still, only four diseases are classified in that case, while our CNN classifies five eye diseases and presents a less complex architecture. Similarly, the EfficientNetB0 model presents an accuracy greater than 1.47% but classifies only four diseases.

It can also be observed that the EfficientNet CNN algorithm [23] presents an accuracy of 94%, and our proposal obtains 3% more accuracy when working with one more class.

The above comparisons suggest that our proposal presents adequate accuracy for classifying five diseases using a less complex architecture.

## 5. Conclusions

Artificial intelligence (AI) is increasingly woven into the fabric of our daily lives, transforming how we interact with technology and the world around us. In healthcare, AI-driven tools can assist in diagnosing diseases, predicting patient outcomes, and personalizing treatment plans, ultimately improving medical care.

In this work, a deep learning algorithm was applied to diagnose eye diseases; in particular four diseases were diagnosed, glaucoma, cataract, retina diseases, and diabetic retinopathy.

The proposed classification model is a custom convolutional neural network that includes three convolutional layers, three max pooling layers, one batch normalization layer, one flattening layer, two hidden layers, and one output layer.

Pre-processing was applied to the images to improve the results obtained by related works. The results were compared between the model that used the filters and the one that did not use pre-processing, and the final result showed that the use of the filters improved the performance of the model.

The following metrics were used to test the performance of the model: confusion matrix, accuracy, recall precision, and F1-score. Hold-out (80-20) was the validation algorithm.

These metrics showed an average of 0.974 for the four eye diseases and a recall of 0.95, indicating that our proposal effectively distinguishes between an image of a normal eye and an image of a sick eye.

This work aimed to present a model for classifying eye diseases that showed adequate performance and was economical. The objective was met, since our proposal showed better results than those works that used only four classes, while our algorithm handled five classes. The software is implemented in a free and open-source language.

In future work, we intend to apply a different pre-processing method to obtain images from which more main features can be extracted. We also have the idea of applying Siamese convolutional networks and analyzing their performance to our proposal.

## Figures and Tables

**Figure 1 diagnostics-15-00916-f001:**
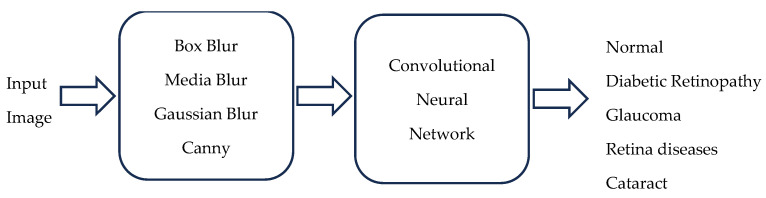
Proposed methodology for the classification of eye diseases.

**Figure 2 diagnostics-15-00916-f002:**
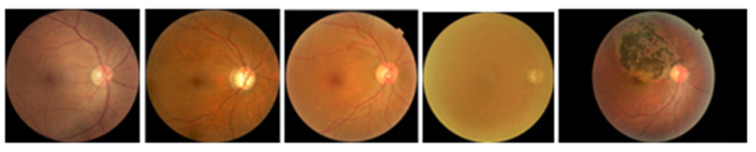
Example of the images in the dataset. Normal, glaucoma, diabetic retinopathy, cataract, and retina disease, respectively.

**Figure 3 diagnostics-15-00916-f003:**
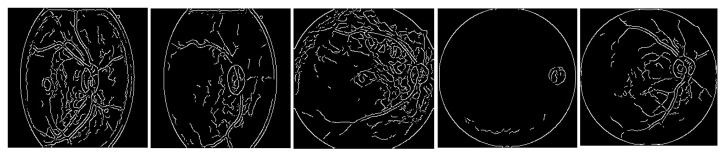
Results from the Canny filter applied to the five eye diseases.

**Figure 4 diagnostics-15-00916-f004:**
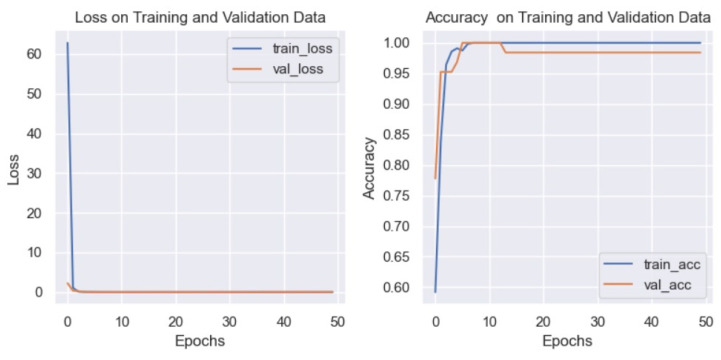
Graph of loss (**left**) and accuracy (**right**) without applying filters to the eye images.

**Figure 5 diagnostics-15-00916-f005:**
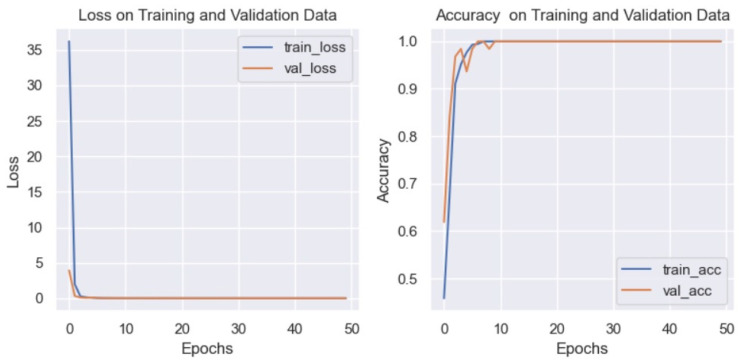
Graph of loss (**left**) and accuracy (**right**) after applying blur and Canny filters to the eye images.

**Figure 6 diagnostics-15-00916-f006:**
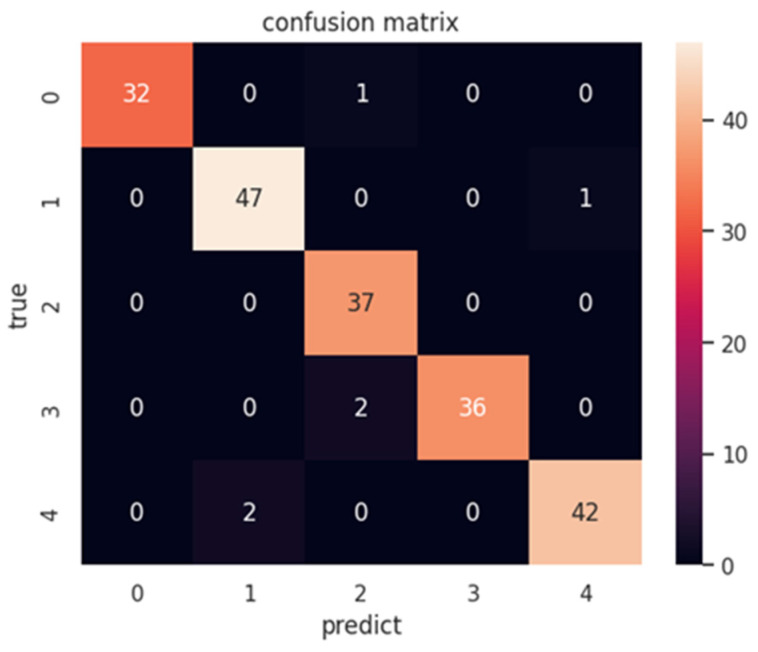
Confusion matrix for the five eye diseases: Class 0 (glaucoma), Class 1 (cataract), Class 2 (retina), Class 3 (diabetic retinopathy), and Class 4 (normal).

**Table 1 diagnostics-15-00916-t001:** Confusion matrix.

True Negatives	False Positives
False Negatives	True Positives

**Table 2 diagnostics-15-00916-t002:** Results of precision: macro, micro, weighted, and none.

Precision	
Precision macro	0.9799
Precision micro	0.97
Precision weighted	0.9712
Precision none	
Glaucoma	1
Cataract	0.9591
Retina	0.925
Diabetic retinopathy	1
Normal	0.9767

**Table 3 diagnostics-15-00916-t003:** Results of accuracy: normalized and non-normalized.

Accuracy	
Normalized accuracy	0.97
Non-normalized Accuracy	194

**Table 4 diagnostics-15-00916-t004:** Results of recall: macro, micro, weighted, and none.

Recall	
Recall macro	0.9701
Recall micro	0.97
Recall weighted	0.97
Recall none	
Glaucoma	0.9696
Cataract	0.9791
Retina	1
Diabetic retinopathy	0.9473
Normal	0.9545

**Table 5 diagnostics-15-00916-t005:** Results of F_1_-score: macro, micro, weighted, and none.

F_1_-Score	
F_1_-score macro	0.9706
F_1_-score micro	0.97
F_1_-score weighted	0.97
F_1_-score none	
Glaucoma	0.9846
Cataract	0.9690
Retina	0.9610
Diabetic retinopathy	0.9729
Normal	0.9655

**Table 6 diagnostics-15-00916-t006:** Comparison of our proposal against related works.

Year	Algorithm	Classes	Metrics
2022	Vision transformer-based approach with three architectures with 8, 14, 24 layers [19]	Age-related macular degeneration, cataracts, diabetes, glaucoma, hypertension, and myopia	With 14 layers F_1_-score = 83.49%, sensitivity = 84% precision= 83%, and Kappa score = 0.802
2023	Single-shot detection, whale optimization algorithm with levy flight and wavelet search strategy, and ShuffleNet V2 model [20]	Glaucoma, cataract, diabetic retinopathy, and normal	Accuracy = 99.1%, precision = 98.9%, recall = 99%, F_1_-Score = 98.9%, Kappa = 96.4%, sensitivity = 98.9%, and specificity = 96.3%.
2023	EfficientNetB0, VGG-16, and VGG-19 models [21]	Glaucoma, cataract, diabetic retinopathy, and normal	With EfficientNetB0 accuracy = 98.47%, precision = 96.98%, recall = 96.91%, and AUC = 99.84%.
2023	CLAHE and convolutional neural network [22]	Glaucoma, cataract, diabetes, age-related macular degeneration, hypertension, and pathological myopia, as well as other diseases that are not specifically mentioned	Experiment 1: multiclass classification accuracy = 60.31% and AUC = 85% Experiment 2: binary classification accuracy between 98% and 100%, recall from 97.99% to 100%, and precision between 96% and 100%
2023	Convolutional neural network and a pre-trained model, EfficientNet CNN [23]	Glaucoma, cataract, diabetic retinopathy, and normal	With EfficientNet CNN accuracy = 94%
2024	VGG-16 convolutional neural network [24]	Normal retina, diabetic macular edema, choroidal neo-vascular membranes, and age-related macular degeneration	Accuracy = 94% and, after fine tuning, it approaches 97%
2024	VGG-16, Xception and MobileNet for feature selection and CNN for classification [25]	Choroidal neovascularization, diabetic macular edema and drusen, and normal	MobileNet with CNN ensemble model accuracy of 95.34%
2024	Fundus-DeepNet system [26]	Normal, diabetic retinopathy, glaucoma, cataracts, AMD, myopia, hypertension, and other abnormalities	F_1_-score = 88.56%, Kappa score = 88.92%, and AUC = 99.76%
2024	Our proposal	Glaucoma, cataract, retina diseases, diabetic retinopathy, and normal	Precision = 97%, accuracy = 97%, recall = 97%, and F_1_-score = 97%

## Data Availability

The data presented in this study are openly available in Kaggle. [Cataract Prediction dataset] [https://www.kaggle.com/datasets/jr2ngb/cataractdataset (accessed on 27 March 2025)].

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
