# Peer review of "Identification of Eye Diseases Through Deep Learning"

_diagnostics, 2025, doi:10.3390/diagnostics15070916_

Round 1
Reviewer 1 Report
Comments and Suggestions for Authors
The article develops a model for the detection of eye diseases with deep learning techniques. In particular, the goal of automating medical diagnosis processes and increasing access to health services is important as an article approach. However, there are deficiencies and points that need to be improved in some sections of the article;
- The Introduction section of the article consists of the disease definition. The advantages of the proposed method and the deficiencies it eliminates in the literature are not mentioned. The contributions of the proposed method to the literature should be given in this section in bullet points. The evaluation of the effectiveness of the method in this section will attract the attention of readers.
- The related studies section has discussed several existing studies in detail, but more literature should be used. It is recommended to include more studies with less detail. The total number of references of 17 is insufficient for this study topic and it is recommended to use at least 30 studies.
- Details about the dataset used should be given. The size, sources and diversity of the dataset should be detailed. Imbalances and class numbers in the dataset should be included.
- The contribution of image processing techniques to the success of the model has not been discussed in detail. The effectiveness of median blur and canny filters should be evaluated.
- The resolution of Figures 1, 2 and 3 should be increased. It is difficult to follow and should be updated. If necessary, disease sections etc. should be marked.
- A detailed description of the 11-layer CNN architecture used should be provided. In particular, more information should be provided about the purpose, parameters and output dimensions of each layer. The hyperparameters of the model should be explained.
- A comparison with Transfer learning, ViT or different CNN architectures that have been proven effective in recent years can strengthen the contribution of the study.
- There is no results section in the article, it should definitely be added and a detailed evaluation should be made. The discussion section can be combined with the Results section. It should be discussed how the model can be used in real clinical scenarios.
- The limitations of the study should be clearly stated. For example, cases where the model misclassified and error analysis should be done.
Author Response
The article develops a model for the detection of eye diseases with deep learning techniques. In particular, the goal of automating medical diagnosis processes and increasing access to health services is important as an article approach. However, there are deficiencies and points that need to be improved in some sections of the article;
- The Introduction section of the article consists of the disease definition. The advantages of the proposed method and the deficiencies it eliminates in the literature are not mentioned. The contributions of the proposed method to the literature should be given in this section in bullet points. The evaluation of the effectiveness of the method in this section will attract the attention of readers.
Res. The Introduction Section has been improved, paragraphs connecting the different parts that make up this section have been added. The definition of Artificial Intelligence and its applications in different areas, especially in the field of medicine, has been added.
- The related studies section has discussed several existing studies in detail, but more literature should be used. It is recommended to include more studies with less detail. The total number of references of 17 is insufficient for this study topic and it is recommended to use at least 30 studies.
Res. 13 references on the applications of Artificial Intelligence in the field of medicine were included
- Details about the dataset used should be given. The size, sources and diversity of the dataset should be detailed. Imbalances and class numbers in the dataset should be included.
Res. From line 271-276, we have the following paragraph that indicates the details of the dataset: Two datasets were used. From the first dataset, called Cataract Prediction [29], the subset Retina_Disease was obtained; this subset does not include diabetic retinopathy. The second dataset was also obtained from Kaggle, and its name is Eye_Diseases_Classification [30]; in this dataset are the subsets: Normal, Cataract, Diabetic Retinopathy, and Glaucoma diseases. Therefore, there are five classes, each with 200 instances, which means the dataset is balanced.
- The contribution of image processing techniques to the success of the model has not been discussed in detail. The effectiveness of median blur and canny filters should be evaluated.
Res. Graphs were included showing the efficiency of the model without the application of the filters and with the use of the filters.
- The resolution of Figures 1, 2 and 3 should be increased. It is difficult to follow and should be updated. If necessary, disease sections etc. should be marked.
Res. Figures 1, 2, and 3 were improved
- A detailed description of the 11-layer CNN architecture used should be provided. In particular, more information should be provided about the purpose, parameters and output dimensions of each layer. The hyperparameters of the model should be explained.
Res. A detailed description of the 11-layer CNN architecture was included
- A comparison with Transfer learning, ViT or different CNN architectures that have been proven effective in recent years can strengthen the contribution of the study.
Res. In Table 6, the work of Vision Transformer-based approach Ref[19] is shown, transfer learning works (VGG-16, Xception and MobileNet) are in Refs. [21], [23-25], and for other types of convolutional networks we have the Fundus-DeepNet system, Ref. [26].
- There is no results section in the article, it should definitely be added and a detailed evaluation should be made. The discussion section can be combined with the Results section. It should be discussed how the model can be used in real clinical scenarios.
Res. In line 308, Section 3, section of Results is shown.
- The limitations of the study should be clearly stated. For example, cases where the model misclassified and error analysis should be done.
Res. A paragraph was added (line 390) explaining, based on the confusion matrix, in which cases our proposal fails.
Reviewer 2 Report
Comments and Suggestions for Authors
I have examined your study titled "Identification of eye diseases through deep learning" in detail. The abstract section was written successfully. The introduction section was weak. Here, the diseases were explained and not connected to anywhere. A paragraph should be added to the artificial intelligence section with these, then the innovative aspects of the article, its contributions to the literature, etc. should be discussed. Finally, this section should be concluded with the organization of the article. I would like to state that there are many deficiencies in this section. So many subheadings are against the nature of the introduction section. Artificial intelligence programs have so many subheadings. When the material and method sections are considered, general information is given. For example, what are the advantages of applying these filters? What would happen if they were not applied? No information is given when it comes to the proposed model section. The layers, parameters, etc. values ​​in this model should be presented. As a result, a subheading called proposed model should be opened in the article and this section should be written in detail. The result section was very weak. Would the performance of your model decrease if you did not use a filter? Different statistical tests and ablation studies should be applied here. If the deficiencies I have stated are eliminated, the quality of the article will increase.
Author Response
I have examined your study titled "Identification of eye diseases through deep learning" in detail. The abstract section was written successfully.
- The introduction section was weak. Here, the diseases were explained and not connected to anywhere.
Res. Paragraphs were included that connect each of the subsections of the Introduction without the need to indicate them.
-A paragraph should be added to the artificial intelligence section with these, then the innovative aspects of the article, its contributions to the literature, etc. should be discussed.
Res. A paragraph was added presenting the definition of Artificial Intelligence and its applications in different areas, especially in the field of medicine.
-Finally, this section should be concluded with the organization of the article. I would like to state that there are many deficiencies in this section. So many subheadings are against the nature of the introduction section.
Res. Document organization was added
Artificial intelligence programs have so many subheadings. When the material and method sections are considered, general information is given. For example, what are the advantages of applying these filters? What would happen if they were not applied?
Res. Graphs were included showing the efficiency of the model without the application of the filters and with the use of the filters.
No information is given when it comes to the proposed model section. The layers, parameters, etc. values ​​in this model should be presented. As a result, a subheading called proposed model should be opened in the article and this section should be written in detail.
The result section was very weak. Would the performance of your model decrease if you did not use a filter? Different statistical tests and ablation studies should be applied here. If the deficiencies I have stated are eliminated, the quality of the article will increase.
Res. Graphs were included showing the efficiency of the model without the application of the filters and with the use of the filters.
Reviewer 3 Report
Comments and Suggestions for Authors The article is generally well organized, although the introductive part “eye diseases” is very superficial and contains some inaccuracies, which have been pointed out below. It could benefit from being rewritten. A “conclusion” section would also be appreciable to offer a synthesis of the results and the discussion to the reader. The introductive portion “1.2 related work” and the table 6 offer a helpful outline of other related works and enable the reader to put this study in context. Citations are appropriate and up to date. The methodology is clearly explained, although the paper would benefit from a brief paragraph to present Canny filters. The data is presented clearly, thus easily understandable and interpretable. Minutiae:- lines 31-32 contain a citation which could be paraphrased. 33-39 are too general and do not add any information. It would be appropriate to develop on the topic of pattern recognitions -line 37- in relation to eye disease diagnosis.
- lines 42-43 incorrect definition of glaucoma.
- line 58 non-proliferative DR. The retina portion could benefit from being rewritten.
- 162: tautology.
- 209: it is table 1
- Lines 226 is quite unclear
Overall, the English level is adequate, even though some portions could benefit from rephrasing to improve understanding and limit uncertainty over its interpretation.
Author Response
The article is generally well organized, although the introductive part “eye diseases” is very superficial and contains some inaccuracies, which have been pointed out below. It could benefit from being rewritten. The introductive portion “1.2 related work” and the table 6 offer a helpful outline of other related works and enable the reader to put this study in context. Citations are appropriate and up to date.
A “conclusion” section would also be appreciable to offer a synthesis of the results and the discussion to the reader.
Res. Section 5 of Conclusions was included
The methodology is clearly explained, although the paper would benefit from a brief paragraph to present Canny filters. The data is presented clearly, thus easily understandable and interpretable.
Res. We added text describing the basic operation of the Canny filter.
Minutiae:
lines 31-32 contain a citation which could be paraphrased.
Res. The citation was removed
33-39 are too general and do not add any information. It would be appropriate to develop on the topic of pattern recognitions -line 37- in relation to eye disease diagnosis.
Res. That part was rewritten
lines 42-43 incorrect definition of glaucoma.
line 58 non-proliferative DR. The retina portion could benefit from being rewritten.
Res. The features of every disease was deleted, the authors consider they are not an important part.
162: tautology.
Res. Corrected
209: it is table 1
Res. Corrected
Lines 226 is quite unclear
Res. Corrected
Round 2
Reviewer 1 Report
Comments and Suggestions for Authors
The authors have completed all suggested corrections completely.
Author Response
Thanks for the comments
Reviewer 2 Report
Comments and Suggestions for Authors
Thanks for the revision. Some deficiencies in the study were addressed in the revision. However, there are still serious deficiencies. What is the innovative aspect of the study? Please add a paragraph under the subheading contrubition and novlty. I cannot find this in the article. Detail the model used in the study. You can provide a summary, you can add an explanatory figure. In the results section, only a confusion matrix related to the result of the model has been added. You need to add some statistical analyses and ablation studies to justify the results section. How did you come to the conclusion that these results are more successful than the CNN-based models or ViTs accepted in the literature? I would like to state that the discussion section is very insufficient. Even adding a table comparing your own results with the literature in this section would bring this section to a better point. The similarity rate of the article is high. Please upload the clean version of the article in addition to the marked version.
Comments on the Quality of English LanguageSpelling and grammatical errors may be reviewed.
Author Response
- Thanks for the revision. Some deficiencies in the study were addressed in the revision. However, there are still serious deficiencies. What is the innovative aspect of the study? Please add a paragraph under the subheading contrubition and novlty. I cannot find this in the article.
Res. The following text was added to the Introduction section and highlighted in yellow.
Our main contribution is an own-designed CNN, with a less complex architecture than a pre-trained algorithm, which allows, firstly, to differentiate a healthy eye from a diseased eye effectively and, secondly, to classify four eye diseases, such as: Cataract, Diabetic Retinopathy, Glaucoma, and other Retina diseases. All this is carried out through pre-processing, using Blur filters and the Canny edge detector, which allows a better detection of main features and, consequently, a adequate performance of the CNN.
- Detail the model used in the study. You can provide a summary, you can add an explanatory figure.
Res. The different stages of CNN are explained in the subsection 2.7 Convolution
The Convolutional Stage shows the following layers.
There are three layers of Convolution:
- Layer 1: 32 filters with a kernel of 5 x 5.
- Layer 2: 64 filters with a kernel of 3 x 3.
- Layer 3: 128 filters with a kernel of 3 x 3.
The MaxPooling layers have kernels with same dimension of 3 x 3.
A Batch Normalization layer is included.
The Flatten layer delivers a vector of dimension 100 to the Multilayer Perceptron.
In the Multilayer Perceptron Stage, we proposed two hidden layers with 100 and 23 neurons, respectively, with ‘relu’ activation function and one output layer with ‘softmax’ activation function. Adam optimizer was applied. The loss function was categorical crossentropy and we proposed 50 epochs and a batch size of 50.
- In the results section, only a confusion matrix related to the result of the model has been added. You need to add some statistical analyses and ablation studies to justify the results section.
Res. Figures 4 and 5 show the loss and accuracy plots without preprocessing and with the use of Bluy and Canny filters, respectively. This corresponds to an ablation study.
- How did you come to the conclusion that these results are more successful than the CNN-based models or ViTs accepted in the literature?
Res. We have not made any comparison with the ViTs models, in fact it is not possible to do so because the work in reference [19] handles six classes, that is, it is a different data set so a direct comparison cannot be made.
In the case of comparison with CNNs, references 20, 21 and 23 handle only four classes, while our proposal handles 5 classes and presents, in some cases, better results. It is worth mentioning that these works use pre-trained CNNs that present more complex architectures.
- I would like to state that the discussion section is very insufficient. Even adding a table comparing your own results with the literature in this section would bring this section to a better point.
Res. Table 6 shows the comparisons between related works and our proposal.
A portion of the Discussion section has also been modified. It is highlighted in yellow.
Reviewer 3 Report
Comments and Suggestions for Authors
At line 45 the definition of glaucoma is incorrect, in the sense that glaucoma is a disease of the optic nerve which leads to progressive blindness. Raised intraocular pressure is a risk factor for its development.
Author Response
At line 45 the definition of glaucoma is incorrect, in the sense that glaucoma is a disease of the optic nerve which leads to progressive blindness. Raised intraocular pressure is a risk factor for its development.
Res. This is the definition of Glaucoma in the paper.
Glaucoma [2] is the term used to define the increase in intraocular pressure, which causes alteration of the optic nerve. The presence of Glaucoma is one of the reasons for vision loss, which spreads through the eye.
Round 3
Reviewer 2 Report
Comments and Suggestions for Authors
Thank you for the revision. Some subheadings have not been updated after the article was edited. Also, spelling and design errors should be reviewed.
Author Response
The spelling and design errors were reviewed